# Water-Soluble *closo*-Docecaborate-Containing Pteroyl Derivatives Targeting Folate Receptor-Positive Tumors for Boron Neutron Capture Therapy

**DOI:** 10.3390/cells9071615

**Published:** 2020-07-03

**Authors:** Fumiko Nakagawa, Hidehisa Kawashima, Taiki Morita, Hiroyuki Nakamura

**Affiliations:** 1School of Life Science and Technology, Tokyo Institute of Technology, 4259 Nagatsuta-cho, Midori-ku, Yokohama 226-8503, Japan; fu-nakagawa@hitachi-chem.co.jp; 2Laboratory for Chemistry and Life Science, Institute of Innovative Research, Tokyo Institute of Technology, 4259 Nagatsuta-cho, Midori-ku, Yokohama 226-8503, Japan; kawashima@ims.tsukuba.ac.jp (H.K.); t.morita@res.titech.ac.jp (T.M.)

**Keywords:** boron neutron capture therapy, folate, FRα, *closo*-dodecaborate, water-soluble

## Abstract

Water-soluble pteroyl-*closo*-dodecaborate conjugates (PBCs 1–4), were developed as folate receptor (FRα) targeting boron carriers for boron neutron capture therapy (BNCT). PBCs 1–4 had adequately low cytotoxicity with IC_50_ values in the range of 1~3 mM toward selected human cancer cells, low enough to use as BNCT boron agents. PBCs 1–3 showed significant cell uptake by FRα positive cells, especially U87MG glioblastoma cells, although the accumulation of PBC 4 was low compared with PBCs 1–3 and L-4-boronophenylalanine (L-BPA). The cellular uptake of PBC 1 and PBC 3 by HeLa cells was arrested by increasing the concentration of folate in the medium, indicating that the major uptake mechanisms of PBC 1–3 are primarily through FRα receptor-mediated endocytosis.

## 1. Introduction

Boron neutron capture therapy (BNCT) has been attracting attention as a noninvasive radiotherapy in cancer treatment. Although the energy of the thermal neutron is extremely low (~0.5 eV), the ^10^B neutron capture reaction produces high linear energy alpha particles (^4^He) and ^7^Li nuclei that dissipate their energy (2.4 MeV) while passing through the cell diameter (approximately 5–9 μm) in tissue, giving a fatal cell-killing effect to cancer cells. Therefore, the combination of boron agents with sufficient and selective accumulation of ^10^B in cancer cells and an appropriate neutron source is essential for successful cancer treatment with BNCT [1,2]. In the last decade, accelerator-based thermal neutron generators for BNCT have been developed worldwide [3,4,5,6,7,8,9,10], and one was approved in Japan this year as a medical device [6,7] in combination with L-4-boronophenylalanine (L-BPA) [11] for the treatment of head and neck carcinoma patients. It is known that L-BPA is actively accumulated into cancer cells through L-type amino acid transporter 1 (LAT-1) [12,13], which is overexpressed in many cancer cells. However, there are still many patients for whom L-BPA is not applicable. One of the reasons for low L-BPA accumulation in some patients’ tumors may be the low expression of LAT-1 in their tumors; thus the development of novel boron carriers applicable to various cancers including L-BPA-negative tumors is required for further development of BNCT. 

Folate is one of the B vitamins and is necessary for the synthesis of purines and thymidine as well as for methylation of DNA, proteins and lipids via *S*-adenosyl methionine [14]; thus it is especially important for frequent cell division [15]. Folate is known to be taken into cells via the folate receptors, cysteine-rich cell-surface glycoproteins [16]. Since the folate receptors are overexpressed on the surface of many cancer cells including HeLa and U-87 MG [17,18], they have attracted attention as targets for cancer treatment [19,20,21]. Various folate receptor-targeted therapies for cancer have been investigated including folate-toxin conjugates, folate-conjugated chemotherapeutic agents, folate receptor-targeted immunotherapy, and folate-conjugated liposomes, micelles, and other nanoparticles [22]. The folate receptor-targeted approach has also been used to develop boron delivery vehicles, such as liposomes [23], polyamidoamine dendrimers [24], boron nanoparticles [25] and nanotubes [26,27], and gold nanoparticles [28], in BNCT. However, there are few reports on low molecular-weight boron agents [29,30]. In all cases, the lipophilic ortho-carborane, as the source of boron, was conjugated with folate, so the boron carriers did not possess adequate water solubility for administration. In fact, 500 mg/kg L-BPA is clinically administrated to achieve the required boron concentration in the tumor [31]. Thus, the water solubility of boron carriers is one of the essential requirements. We focused on a *closo*-dodecaborate, a water soluble and low toxicity boron cluster, and introduced it into folate. It is known that the pteroyl group of folate is essential for the interaction with folate receptors [32,33]. In this paper, we synthesized pteroyl-*closo*-dodecaborate conjugates (PBCs) and examined their cell uptake using folate receptor (FRα) positive and negative cells.

## 2. Materials and Methods

### 2.1. General Methods

Boron concentrations were measured with inductively coupled plasma optical emission spectroscopy (ICP-OES) (Thermo Fisher Scientific Inc. Waltham, MA, USA). The statistical significance of the results was analyzed using the Student’s t-test for unpaired observations and Dunnett’s test for multiple comparisons. All protocols for in vivo study were approved by the Institutional Animal Care and Use Committee of Tokyo Institute of Technology (D2015011). (Et_3_NH)_2_[B_12_H_12_] was purchased from Katchem Ltd. (Prague, Czech Republic), and Na_2_[^10^B_12_H_12_] was kindly provided by Stella Chemifa Co. (Osaka, Japan). Other chemicals were purchased from Tokyo Chemical Industry Co., Ltd. (TCI). HeLa (human cervix epithelioid carcinoma) and A549 (human alveolar adenocarcinoma) cells were kindly provided by the Cell Resource Center for Biomedical Research, Institute of Development, Aging and Cancer, Tohoku University. U-87 MG (human malignant glioma) cells were purchased from Cosmo Bio Co., Ltd.

### 2.2. MTT Assay

HeLa, U-87 MG, and A549 were seeded in 96-well plates in RPMI-1640 medium (Wako Pure Chemicals) supplemented with 10% fetal bovine serum and 1% penicillin/streptomycin (Gibco; Thermo Fisher Scientific, Waltham, MA, USA) at the density of 5 × 10^3^ cells/well. After 24 h of cell attachment, the cells were exposed to PBC 1, PBC 2, PBC 3, PBC 4, and L-BPA-fructose complex at final concentrations ranging from 0.1 to 10 mM for 72 h at 37 °C. At the end of the incubation period, the mitochondrial function was verified with 0.5 mg/mL MTT (3-(4,5-dimethylthiazol-2-yl)-2,5- diphenyltetrazolium bromide) for 2 h at 37 °C and quantified spectrophotometrically at 595 nm by Biorad microplate reader. Data were expressed as a percentage of the viability of each control.

### 2.3. Cell Uptake Study

PBCs or L-BPA solutions were prepared from PBCs (1500–3000 ppm B dissolved in ultrapure water (Milli-Q water)) or L-BPA (2400 ppm B in the fructose solution) and medium. HeLa, U-87 MG, and A549 cells were cultured at the density of 3 × 10^5^ cells/dish (U-87 MG) or 1 × 10^6^ cells/dish (other cells) in 6-well plates in the medium (1mL) at 37 °C in a 5% CO_2_ incubator for 24 h; then the medium was removed and replaced with the drug solutions for 1, 3, and 12 h. This medium was removed and the cells were washed three times with PBS (phosphate-buffered saline), collected by rubber, and dissolved in a mixed solvent of 60% HClO_4_ and 30% H_2_O_2_ (1:2 *v/v*) at 70 °C for 2 h, and then Milli-Q water was added up to 5 mL total. After filtering through a membrane filter (0.5 µm*φ,* 13JP050AN, ADVANTEC, Japan), the boron concentration of the resulting solutions was measured by ICP-OES.

### 2.4. Immunofluorescence

HeLa cells were plated into *φ*35 dishes (1 × 10^4^ cells) including the glass coverslips (10 mm square) and incubated at 37 °C for 24 h. After PBC 1 (100 ppm B) treatment for 3 h, the cells were washed three times with PBS and fixed with 4% paraformaldehyde in PBS for 10 min. After washing with 0.4 % Triton X-100 in PBS for 5 min, the cells were incubated with anti-BSH (mercaptoundecahydro-*closo*-dodecaborate) antibody at 4 °C overnight. After washing three times with PBS, the cells were incubated with the HRP (horseradish peroxidase)-conjugated secondary antibody for 2 h, the cells were washed three times with PBS, and tyramide-Cy3 was added. After incubating for 5 min, the cells were washed three times with PBS, and DAPI (4’,6-diamidino-2-phenylindole) solution was added. After incubating for 5 min, the cells were washed three times with PBS and mounted with Prolong Gold anti-fade reagent (invitrogen). Fluorescence images were analyzed using a confocal laser scanning microscope (Carl Zeiss, LSM780). 

## 3. Results

### 3.1. Design and Synthesis of Pteroyl Closo-Dodecaborates

Although the synthesis of differentially functionalized folate derivatives is not an easy task, Fuchs et al., have developed effective protocols through pteroyl azide [34]. Thus, we designed PBCs 1–4 from pteroyl azide and *closo*-dodecaborate **1** [35] conjugated with amino acid linkers (Figure 1).

Synthesis of amino acid linker-conjugated *closo*-dodecaborates is shown in Scheme 1. According to the reported procedure with modification [36], *closo*-dodecaborate **1** was synthesized from bis-tetrabutylammonium form **2**, which was easily prepared from the commercially available *closo*-dodecaborate bis-sodium form, through the dioxane complex **3**. Then *closo*-dodecaborate **1** was treated with 1-benzyl-*N*-*tert*-butoxycarbonyl-L-glutamate using ethyl-3-(3-dimethylaminopropyl)carbodiimide (EDCI), *N,N*-dimethylaminopyridine (DMAP), and triethylamine (TEA) to give compound **4** in 68% yield. Hydrogenesis to remove the benzyl protective group followed by acid hydrolysis afforded compound **5** in 71% yield. Similarly, glycine and alanine linked *closo*-dodecaborates **6** and **8** were prepared from compound **1** with *N-tert*-butoxycarbonyl-L-glycine and *N-tert*-butoxycarbonyl-L-alanine, respectively, and the acid hydrolysis to remove the Boc protection gave compounds **7** and **9**, quantitatively.

PBCs 1–4 were synthesized as shown in Scheme 2. First, pteroyl azide was synthesized from folate through compound **10** according to the reported procedures [31] and conjugated with *closo*-dodecaborates **1**, **5**, **7**, and **9** in the presence of 1,8-diazabicyclo[5.4.0]undec-7-ene (DBU). The resulting TBA forms of pteroyl derivatives were converted to bis-tetramethylammonium forms followed by bis-sodium forms to afford PBCs 1–4. The chemical structures of PBCs 1–4 were identified by ^1^H, ^13^C, and ^11^B NMR, high resolution mass spectroscopy, and IR (see the Appendix A).

### 3.2. Cytotoxicity of PBCs

As mentioned earlier, a high dose is necessary to achieve the required boron concentration in the tumor for BNCT. Thus, not only adequate water solubility but also low cytotoxicity is essential for BNCT boron agents. We first examined the cytotoxicity of synthesized PBCs toward three human cancer cell lines using MTT assay: HeLa (human cervical carcinoma) and U87MG (human glioblastoma) cells are FRα positive and A549 (human alveolar adenocarcinoma) cells are FRα negative (see Appendix A). L-BPA was used as a positive control. The results are summarized in Table 1. PBCs 1–4 exhibited IC_50_ values (the concentrations required for 50% inhibition) in a range of 1–3 mM toward these human cancer cells, indicating that PBCs 1–4 have adequately low cytotoxity, enough to use as BNCT boron agents.

### 3.3. Cellular Uptake and Distribution of PBCs

We next examined the boron accumulation of PBCs 1–4 in these three cell lines. The results are shown in Figure 2. Both PBC 1 and PBC 2 were similarly accumulated into HeLa cells, whereas twice the boron accumulation was observed in the case of PBC 3 and L-BPA. To our surprise, the cell uptake of PBC 4 was quite low compared with PBCs 1–3. A similar tendency was observed in the accumulation into U87MG cells. Interestingly, the boron accumulation of PBCs 1–3 was higher than that of L-BPA. It is known that L-BPA is accumulated into cells through LAT1 and that the expression level of LAT1 is relatively low on the surface of U87MG cells. Therefore, the significant accumulation difference between PBCs 1–3 and L-BPA is probably due to the FRα positive and LAT1 negative property of U87MG cells. In fact, a lower accumulation of PBCs into FRα negative A549 cells was observed compared with those of FRα positive HeLa and U87MG cells.

It should be noted that PBC 3 always exhibited significant accumulation into the cells. Therefore, we carried out folate concentration-dependent boron accumulation of PBC 1 and PBC 3 into HeLa cells to confirm whether PBCs were accumulated through FRα or not. The results are shown in Figure 3A. Folate concentration-dependent decreases of boron concentration in HeLa cells were observed in the cases of PBC 1 and PBC 3; however, the boron accumulation of L-BPA was not affected by the presence or absence of folate. We also examined the cellular distribution of PBC 1 in HeLa cells by immunostaining using anti-*closo*-dodecaborate antibody. As shown in Figure 3B, PBC 1 was located in the cytosol where FRα normally resides. These results indicate that the major uptake mechanism of both PBC 1 and PBC 3 is primarily through FRα receptor-mediated endocytosis.

## 4. Discussion

For targeting FRα, the lipophilic ortho-carborane-conjugated folate derivatives have been reported so far, as boron carriers of BNCT. However, these boron carriers did not possess adequate water solubility for administration due to the highly lipophilic property of ortho-carborane. In the present study, we have prepared pteroyl-*closo*-dodecaborate conjugates, PBCs 1–4 as FRα targeting boron carriers. By using closo-dodecaborate, instead of lipophilic *ortho*-carborane, PBCs 1–4 showed adequate water-solubility as expected. Combined with low cytotoxicity of PBCs 1–4 which was indicated by MTT assay, these compounds seem to have appropriate properties as BNCT boron agents. In fact, as shown in Figure 2, the cellular uptake study demonstrated significant accumulation of PBCs 1–3 into FRα positive cells, though PBC-4 showed much lower accumulation than PBCs 1–3. The difference between PBC 3 and PBC 4 is a methyl group substituted at the α-position of the amide bond in PBCs. These results suggest that the structure of the amino acid linker is highly important for the property of PBCs. Although the studies on the folate concentration-dependent boron accumulation demonstrated that PBC 1 and PBC 3 were accumulated through FRα receptor-mediated endocytosis, PBC 3 still showed high accumulation into FRα negative A549 cells. This result would indicate that PBC 3 tends to accumulate into cells because of their relatively lipophilic glycine linker. Indeed, as we observed the precipitation of PBC 3 during the biodistribution study (date not shown), even a slight structural modification would have a drastic impact. We believe that further investigation on the structure of the linker moiety will lead to sufficient boron accumulation for BNCT treatment. It should be noted that folate concentration-dependent decreases of boron concentration in HeLa cells were observed in the cases of PBC 1 and PBC 3 as shown in Figure 3A. However, the boron accumulation of L-BPA was not affected by the presence or absence of folate. These results indicate that both PBCs and L-BPA have different uptake pathways. It is known that L-BPA accumulates into tumor cells primarily via the LAT-1 pathway; thus, simultaneous administration of both PBCs and L-BPA can enhance boron uptake by tumor cells. In fact, the survival time of brain tumor model rats treated with both PBC 1 and L-BPA was significantly prolonged in comparison with those treated with PBC 1 or L-BPA alone. [37] 

## 5. Conclusions

We designed and synthesized water-soluble pteroyl-*closo*-dodecaborate conjugates, PBCs 1–4, as FRα targeting boron carriers. PBCs 1–4 have adequately low cytotoxicity with IC_50_ values in the range of 1~3 mM toward these human cancer cells, low enough to use as BNCT boron agents. PBCs 1–3 showed significant cell uptake by FRα positive cells, especially U87MG glioblastoma cells, although the accumulation of PBC 4 was low compared with PBCs 1–3 and L-BPA. The cellular uptake of PBC 1 and PBC 3 by HeLa cells was arrested by increasing the concentration of folate in the medium, indicating that the major uptake mechanisms of PBCs 1–3 are primarily through FRα receptor-mediated endocytosis. Therefore, newly synthesized PBCs 1–3 have the potential to be alternative boron agents that could be applied to various cancers including L-BPA-negative tumors for the further development of BNCT.

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
