# Peer review of "Water-Soluble closo-Docecaborate-Containing Pteroyl Derivatives Targeting Folate Receptor-Positive Tumors for Boron Neutron Capture Therapy"

_cells, 2020, doi:10.3390/cells9071615_

Round 1

Reviewer 1 Report

The paper is an interesting work about the efficiency of new boron carriers to target the tumors in the perspective of BNCT.

The authors can improve the manuscript following these remarks:

  • The abstract needs a deep revision. It summarizes the methodology but without explaining the aim of the work and the main conclusions. The abbreviations used in the abstract are not explained (PBC1_4, FR…).
  • The introduction explains well the context of the research and the strategy. The authors claimed that LBPA is not applicable in all tumors. It would be helpful to explain which ones and also how this work can help regarding this issue.
  • The figure 1 comes too early and would be better within the results part to explain in details the strategy of the synthesis.
  • The providers are not always mentioned (cells, chemicals…)
  • Cellular experiments : RPMI medium is mentioned but HeLa should be cultured in MEM or DMEM
  • For MTT the concentration of PCB is in mM and in ppm in uptake experiment, it could be confusing.
  • Cytotoxicity of PBC (table 1): It is explained that PBCs have a low cytotoxicity. Could you indicate what is the minimum required to consider it as low toxic ?
  • Cellular uptake and distribution of PBCs (Fig 2). Could you comment/discuss some points : For A549 the uptake of PBC3 is similar to LBPA, while it is inferior for PBC 1/2/4 ; for HeLa the uptake of PBC3 is similar to LBPA also ; It is superior only for U87-MG.
  • Fig 3 : The results without Folate should be similar than those of Fig 2 (Hela) but they are very different. Could you explain this?
  • In the in vivo experiment, the cells used for tumor bearing mice are different from the in vitro experiment. Could you explain this choice and the link between in vitro and in vivo experiments ?

Author Response

# Referee 1

The paper is an interesting work about the efficiency of new boron carriers to target the tumors in the perspective of BNCT.

The authors can improve the manuscript following these remarks:

  • The abstract needs a deep revision. It summarizes the methodology but without explaining the aim of the work and the main conclusions. The abbreviations used in the abstract are not explained (PBC1_4, FR…).

<Author Reply> We checked all the abbreviations in the paper and added them appropriately.

  • The introduction explains well the context of the research and the strategy. The authors claimed that LBPA is not applicable in all tumors. It would be helpful to explain which ones and also how this work can help regarding this issue.

<Author Reply> Because still many potential patients for BNCT treatment have negative response to L-BPA. One of the reasons for the low L-BPA accumulation in the patient's tumor may be the low expression of LAT-1 in the tumor. Therefore, development of novel boron agents which can be accumulated through other than LAT-1 is highly important to apply BNCT to those patients. The sentence in the paper is also corrected.

  • The figure 1 comes too early and would be better within the results part to explain in details the strategy of the synthesis.

<Author Reply> Figure 1 was moved to results part, in the beginning of chapter 3.

  • The providers are not always mentioned (cells, chemicals…)

<Author Reply> The providers’ information was included in the manuscript.

  • Cellular experiments : RPMI medium is mentioned but HeLa should be cultured in MEM or DMEM

<Author Reply> According to published literatures, HeLa cells were also cultured in RPMI medium. For example, please see Cell Mol Biol (Noisy-le-grand). 2019 Apr 30; 65(4): 79-82 or PLOS ONE 2014, 9, e87223.

  • For MTT the concentration of PCB is in mM and in ppm in uptake experiment, it could be confusing.

<Author Reply> To describe the cytotoxicity of PBCs, we need the concentration of those compounds in mM. In uptake studies, however, the boron concentration should be described to compare PBCs with L-BPA as boron agents for BNCT.

  • Cytotoxicity of PBC (table 1): It is explained that PBCs have a low cytotoxicity. Could you indicate what is the minimum required to consider it as low toxic ?

<Author Reply> Given the requirements for boron accumulation for BNCT treatment, the IC50 of boron agents should be higher than 0.1 mM.

  • Cellular uptake and distribution of PBCs (Fig 2). Could you comment/discuss some points : For A549 the uptake of PBC3 is similar to LBPA, while it is inferior for PBC 1/2/4 ; for HeLa the uptake of PBC3 is similar to LBPA also ; It is superior only for U87-MG.

<Author Reply> Based on cellular uptake studies, PBC3 would be mainly accumulated through FRa receptor-mediated endocytosis. In addition, we speculate that PBC3 would tend to accumulate into cells because of their relatively lipophilic glycine linker. This lipophilicity would enable high accumulation even into A549, FRa-negative cells.

  • Fig 3 : The results without Folate should be similar than those of Fig 2 (Hela) but they are very different. Could you explain this?

<Author Reply> The difference indicated in the question would be about L-BPA. Because the concentration of L-BPA is different in each experiments. In the experiment shown in Figure 2, the boron concentration is 100 ppm, but it is 25 ppm in Figure 3.

  • In the in vivo experiment, the cells used for tumor bearing mice are different from the in vitro experiment. Could you explain this choice and the link between in vitro and in vivo experiments ?

<Author Reply> Because we have well experienced in vivo experiments using colon 26 tumor-bearing mice (for example, see: J. Control. Release, 2016, 237, 160–167), those mice was chosen. In addition, overexpression of FRa in CT26 cells was indicated by western blot analysis as shown in Figure S1. Combined with the FRa-mediated accumulation of PBCs into FRa-positive cells suggested by in vitro experiments, it would be possible to expect uptake of PBCs into colon 26 tumor-bearing mice through FRa.

Reviewer 2 Report

Title: appropriate and expressive of the content of the manuscript

Introduction: Folate receptors are over-expressed on the surface of many cancer cells, including -------- -->(give a few examples).           Line 52: The 500mg/kg L-BPA -------> Ref ??                                       In general, the introduction is informative and current enough. It could be even better with a little more updated references.

Materials & Methods:                                                                           In general, the methods were appropriately chosen and written. However,                                                                              Section 2.5 In vivo Bio-distribution raises some concern:                    - how many mice, and why females?; how many of the total were used as controls? Were the control mice given the L-BPA-fructose solution?: The tissue sampling -- state the size of each tissue/organ taken, and the amount of blood per sampling. The mice were anesthetized every hour for six hours for cardiac venipunture (blood sampling)? And no stress induced? 

Results:                                                                                         Well-written; with illustrative Table and Figures

Concern: p-values (P>?) must be indicated/written in all cases where differences in data/results were indicated. You used different substances/drugs and or on different cancer cells and got different results at different time intervals.

Discussions & Conclusions:                                                             Adequately written and reflective of result.  

Author Response

# Referee 2

Title: appropriate and expressive of the content of the manuscript

  • Introduction: Folate receptors are over-expressed on the surface of many cancer cells, including -------- -->(give a few examples).

 <Author Reply> We corrected as "including HeLa, U-87 MG"

  •  Line 52: The 500mg/kg L-BPA -------> Ref ?? 

 <Author Reply> The reference was cited in reference 31.

  •  In general, the introduction is informative and current enough. It could be even better with a little more updated references.

 <Author Reply> In the revised manuscript, ref. 10 and ref. 21 were added as recently published papers.

Materials & Methods:                                                                         

 In general, the methods were appropriately chosen and written.

However, Section 2.5 In vivo Bio-distribution raises some concern:                    -

  • how many mice, and why females?

 <Author Reply> Total 27 mice were used. They are classified into 3 types: 1) L-BPA, 2) PBC1, and 3) PBC3. Three mice were sacrificed at 1.5, 3, and 6 h after administration. In general, female mice with 6 weeks old are widely used in in vivo experiments.

  • how many of the total were used as controls?

 <Author Reply> We used three mice as control for the background measurement.

  • Were the control mice given the L-BPA-fructose solution?

 <Author Reply> Yes, as mentioned in manuscript, L-BPA-fructose solution was used.

  • The tissue sampling -- state the size of each tissue/organ taken, and the amount of blood per sampling.

 <Author Reply> The average weight of each organ is as follows: Tumor 0.42 ± 0.11 g; blood 0.58 ± 0.15 g, liver 1.04 ± 0.11 g; kidney 0.28 ± 0.03 g; spleen 0.11 ± 0.02 g; brain 0.34 ± 0.07 g. This information was included in the experimental section.

  • The mice were anesthetized every hour for six hours for cardiac venipunture (blood sampling)? And no stress induced? 

 <Author Reply> Time intervals were corrected as “At selected time intervals (1.5, 3, 6 h)”. Given that the mice were sacrificed right after only once anesthesia, there would not be any induction of stress.

Results:   Well-written; with illustrative Table and Figures

  • Concern: p-values (P>?) must be indicated/written in all cases where differences in data/results were indicated. You used different substances/drugs and or on different cancer cells and got different results at different time intervals.

 <Author Reply> As the reviewer pointed, the statistical significance is especially important in Figures 3A and 4. Therefore, we added “Statistical significance: *P < 0.005 and **P < 0.001 compared with controls (no folate)” in Figure 3A. In addition, P values for tumor and blood were indicated in the text.

Reviewer 3 Report

The presented study covers extremely relevant and promising topics. The idea is interesting, and its originality is well founded in the introduction. In vitro studies have been performed well and the results obtained do not raise my any questions. At the same time, the presented results of the in vivo accumulation study of new compounds seemed to me technically incorrect and do not allow to make any conclusion about the properties of the new substances:

  1. Very small differences between time points, which looks especially strange in the blood.
  2. The lack of accumulation in the tumor, which in itself does not allow to draw any conclusions.
  3. The lack of comparison of boron concentration in the tumor and in the tissues surrounding it, which is key to the potential effectiveness of therapy.

The creation of new substances with high accumulation in tumor tissue through FRα receptors is the main task of the presented study. So, unfortunately, the work can only be published after the redesign of in vivo studies and have to be rejected in present form.

Minor points:

Abbreviations (BNCT, L-BPA, FRalfa) are better to decipher already in the abstract. Otherwise, the abstract becomes unclear without reading the article.

Line 83: "cells were washed with 3X PBS". I think you mean "cells were three times washed with PBS".

Author Response

# Referee 3

The presented study covers extremely relevant and promising topics. The idea is interesting, and its originality is well founded in the introduction. In vitro studies have been performed well and the results obtained do not raise my any questions. At the same time, the presented results of the in vivo accumulation study of new compounds seemed to me technically incorrect and do not allow to make any conclusion about the properties of the new substances:

  1. Very small differences between time points, which looks especially strange in the blood.
  2. The lack of accumulation in the tumor, which in itself does not allow to draw any conclusions.
  3. The lack of comparison of boron concentration in the tumor and in the tissues surrounding it, which is key to the potential effectiveness of therapy.

The creation of new substances with high accumulation in tumor tissue through FRα receptors is the main task of the presented study. So, unfortunately, the work can only be published after the redesign of in vivo studies and have to be rejected in present form.

 <Author Reply> As the reviewer pointed, PBCs developed in the manuscript are still required to be modified in order to obtain in vivo efficacy for BNCT. Therefore, we described “the improvement of PBCs in the blood circulation is necessary to achieve sufficient boron accumulation as an alternative boron agent for BNCT”. However, this is the first to report the boron containing water-soluble folate derivatives as BNCT agents. We strongly believe that PBCs will be more efficient in vivo if we can improve their blood circulation property.  

Minor points:

Abbreviations (BNCT, L-BPA, FRalfa) are better to decipher already in the abstract. Otherwise, the abstract becomes unclear without reading the article.

 <Author Reply> The abbreviations for those words are described in abstract.

Line 83: "cells were washed with 3X PBS". I think you mean "cells were three times washed with PBS".

 <Author Reply> The sentence was corrected.

Round 2

Reviewer 3 Report

I completely agree with the authors of the manuscript that the data on the biological properties of the new water-soluble substance for BNCT deserve publication. But the extremely small differences in the boron concentration after the injection of a positive control (L-BPA) between the maximum (30 min) and minimum points (6 h) in the blood leads to doubts about the technical correctness of in vivo studies. For example, the dynamics of the concentration of L-BPA in the blood plasma and its accumulation parameters are dramatically differ in earlier studies (https://cancerres.aacrjournals.org/content/69/5/2126.long; https://pubmed.ncbi.nlm.nih.gov/8327732/ and others). It also raises the question of why the concentration of boron was not analyzed in normal tissue surrounding the tumor, since it is the key ratio, as well as the tumor/blood ratio.
In my opinion, authors should compare their results with earlier studies using L-BPA as a link. It is also necessary to explain why they considered the ratio of boron concentration between normal and tumor tissues not important.
As an alternative, given the obvious scientific significance of new substances and in vitro studies, it is possible at this stage to completely remove the in vivo block from manuscript, leaving it for experimental revision in the next manuscript.

Author Response

The reviewers' suggestion: As an alternative, given the obvious scientific significance of new substances and in vitro studies, it is possible at this stage to completely remove the in vivo block from manuscript, leaving it for experimental revision in the next manuscript.

<The authors' reply>  According to the reviewers' suggestion, we removed the in vivo experimental part from the manuscript.